# Semantic Diffusion Network for Semantic Segmentation

**Haoru Tan**[*]       **Sitong Wu**[*†]       **Jimin Pi**

Baidu Research
tanhr2014@163.com, wusitong98@gmail.com
jpi@connect.ust.hk

## Abstract

Precise and accurate predictions over boundary areas are essential for semantic segmentation. However, the commonly-used convolutional operators tend to smooth and blur local detail cues, making it difficult for deep models to generate accurate boundary predictions. In this paper, we introduce an operator-level approach to enhance semantic boundary awareness, so as to improve the prediction of the deep semantic segmentation model. Specifically, we first formulate the boundary feature enhancement as an anisotropic diffusion process. We then propose a novel learnable approach called semantic diffusion network (SDN) to approximate the diffusion process, which contains a parameterized semantic difference convolution operator followed by a feature fusion module. Our SDN aims to construct a differentiable mapping from the original feature to the inter-class boundary-enhanced feature. The proposed SDN is an efficient and flexible module that can be easily plugged into existing encoder-decoder segmentation models. Extensive experiments show that our approach can achieve consistent improvements over several typical and state-of-the-art segmentation baseline models on challenging public benchmarks.

## 1   Introduction

Semantic segmentation aims at assigning pixel-wise semantic labels to an image. It plays an important role in various applications such as autonomous driving [9, 13], medical diagnostics [31] and virtual reality [4]. A key challenge in semantic segmentation is how to improve the boundary quality of segmentation results (Figure 1). In general, people have been working on this problem in three different ways: (i) adopting a post-processing module to refine the boundary [26, 23, 43, 39, 58], (ii) jointly learned the edge detection and semantic segmentation task for boundary-awareness [60, 7, 30], and (iii) making training more focused on the boundary by designing a boundary-aware loss with higher sensitivity to the boundary changes [3, 20, 45].

Nevertheless, all these methods explored the boundary information via either post-processing refinement or additional supervision, rather than directly address the intrinsic reason for fuzzy boundary and detail degradation, that is, *vanilla convolution [24] naturally tends to smooth the feature [41, 53–56], making it difficult to capture the boundary cues* (see Figure 2b). Although traditional gradient operators [19, 22] and difference convolutions [53–56, 41] have the ability to perceive boundary, they are sensitive to all the boundaries including intra-class texture edges, which makes it not suitable for semantic segmentation. This motivates us to explore a segmentation-friendly operator with only inter-class boundary awareness to improve the boundary ambiguity in semantic segmentation.

---

[*]Contribute equally
[†]Corresponding author

36th Conference on Neural Information Processing Systems (NeurIPS 2022).

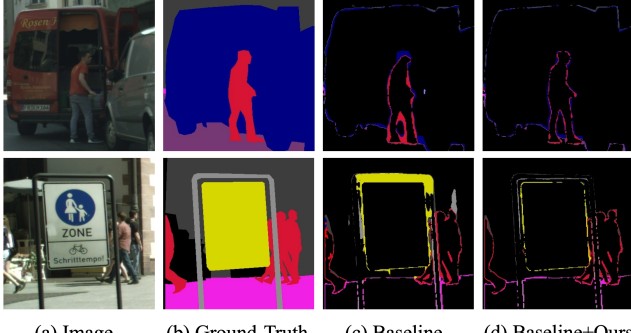

| (a) Image | (b) Ground-Truth | (c) Baseline | (d) Baseline+Ours |

Figure 1: Qualitative analysis of the segmentation error maps. The baseline denotes OCRNet [57], and the non-black pixels indicate the wrong predictions. It can be seen that segmentation error is particularly obvious in the boundary region. Our method achieves significant boundary improvements over the baseline model. The examples are selected and cropped from Cityscapes-*val* [9].

**Our method.** In this paper, we are committed to improving the boundary quality of semantic segmentation from the operator level. Inspired by the anisotropic diffusion [33] in the signal processing field, we formulate the inter-class boundary enhancement as a modified anisotropic diffusion process guided by semantics, which can be solved by the finite difference method. While such a traditional solver can hardly be integrated into deep segmentation networks due to the hyper-parameter sensitivity, numerical instability, and high computational complexity caused by its iterative formulation. To solve this problem, we propose a learnable semantic diffusion network (SDN) to approximate the diffusion process, which parameterizes the traditional solver and only requires only one forward instead of multiple iterations. The SDN aims to construct a differentiable mapping from the original feature to the inter-class boundary-enhanced feature, which consists of a semantic difference convolution and a feature fusion operation. Our proposed SDN is an efficient and flexible module that can be easily plugged into the existing encoder-decoder segmentation models. To evaluate its effectiveness, we integrate our SDN into multiple baseline segmentation models and conduct experiments on two challenging benchmarks (*i.e.,* ADE20K [62] and Cityscapes [9]). Extensive experiments show that our approach can bring consistent improvements for existing segmentation methods with few additional computational costs. In particular, our SDN can bring +2.57% and +1.43% mIoU improvements for Segmenter [40] with ViT-B [11] backbone on ADE20K [62] and Cityscapes [9], respectively. Qualitative visualization verifies the obvious boundary quality improvement of our approach.

## 2 Related Work

**Semantic Segmentation.** In 2015, FCN [18] first proposed to use a fully convolutional network to generate dense predictions, which is regarded as a milestone for semantic segmentation. Since then, lots of works have been exploring larger receptive fields for better scene understanding based on CNN architecture. For example, PSPNet [15] proposed a general pyramid pooling module to fuse features at different scales. DeepLab family [26, 27, 25, 28] used atrous convolution to enlarge the receptive fields while keeping the resolution. Recent works focused on how to capture more precise context [57, 17, 59, 29, 52] and better boundary [58, 3]. Considering that the attention operation [46] has a large receptive field and can adaptively aggregate features, [12, 16, 63, 2] employed the attention-based modules on the top of CNN segmentation networks to adaptively combine local features with their global dependencies. Inspired by the success of Transformer [44] in NLP tasks, SETR [61] and DPT [35] attempted to use Vision Transformer [11] as the backbone. They demonstrated the effectiveness of global receptive fields for semantic segmentation tasks. Inspired by [5], Tran2Seg [51] explored a hybrid encoder and prototype-based transformer decoder. FTN [49] and Segmenter [40] further designed fully transformer networks. SegFormer [50] proved that a lightweight MLP decoder can achieve satisfactory results under a strong transformer encoder. Recently, MaskFormer [8] proposed a simple but effective mask classification scheme to predict a set of binary masks, which unified the semantic, instance, and panoptic segmentation.

**Boundary-aware Segmentation.** Methods for promoting boundary quality in semantic segmentation can be divided into three categories: (1) *Adopting the post-processing module.* Most methods applied a post-processing module on the top of coarse segmentation results to refine the boundary and recover the details [26, 23, 43, 39, 58]. DenseCRF model [23] is the most well-known post-processing method, which achieves fine boundary results by encouraging assigning neighboring pixels with similar colors

to the same semantic label. SegFix [58] proposed a model-agnostic boundary refinement model, which aims to replace the ambiguous predictions at boundaries with predictions at the intra-class areas. (2) *Auxiliary branch for boundary-awareness.* [60, 7, 30] utilized the multi-task learning strategy by jointly learning boundary detection and semantic segmentation together. For example, RPCNet [60] introduced the auxiliary branch for boundary enhancement to the semantic segmentation model. (3) *Boundary-aware loss.* These works guide the model to focus more on boundary accuracy by designing the boundary-aware loss function with high sensitivity to the boundary changes [3, 20, 45]. Inverseform [3] introduced a boundary distance-based measure into the popular segmentation loss functions. Kervadec *et.al.*[20] proposed a boundary loss taking the form of a distance metric on the space of contours for highly unbalanced situations. ABL [45] designed an active boundary loss function for progressively encouraging the predicted boundaries matching the ground-truth boundaries. *Instead of imposing various boundary-related constraints, we consider the intrinsic reason for fuzzy boundary from the operator level, and design a boundary-sensitive operator to make the network naturally capable of modeling and perceiving boundaries.*

**Related Operators.** The convolution operator, which calculates the kernel's response to each local patch of the input feature maps, is not good at modeling boundary information. To solve this, people try to combine the traditional image gradient operator with convolution operators. Local binary convolution (LBC) [19] used a series of pre-defined binary filters to replace learnable kernels in convolution. Subsequently, a series of difference convolution operators represented by central difference convolution (CDC) [53–56] used learnable kernels to capture useful boundary information on the central difference map. PDC [41] further extended the flexibility of CDC and achieved excellent results in boundary detection tasks. In addition, the local self-attention (LSA) [34] operator replaces the fixed kernel with the content-adaptive attentive weights to achieve local feature fusion, making it good at capturing contextual dependence. But it is still not good at perceiving the boundary. *Different from existing methods, we are committed to designing a novel and efficient operator-level solution with a high sensitivity to the inter-class boundaries rather than all the edges.*

## 3 Preliminaries

### 3.1 Diffusion Process

Diffusion is a physical model aimed at minimizing the spatial concentration difference [37], which is widely used in image processing [33, 6, 48, 47]. Given an image tensor $U$ to be smoothed, this process needs to solve the well-known second-order partial differential equation:

$$\frac{\partial U}{\partial t} = Div(D \cdot \nabla U), \tag{1}$$

where $\frac{\partial U}{\partial t}$ is the time derivative of the solution tensor, $Div$ is the divergence operator, $D$ is the diffusivity tensor, which describes the speed of the diffusion process, $\nabla$ is the gradient operator. With increasing time $t$, the solution $U_t$ of this process will correspond to increasingly smoothed versions of the original tensor.

The concrete form of $D$ determines the properties of the PDE. For $D = 1$, it is called homogeneous linear diffusion, where the diffusion velocity of spatial concentration is exactly the same in all directions. For a spatial-dependent $D = D(x)$ (*e.g.* learnable kernel), the process is linear and inhomogeneous. However, the two kinds of simple linear diffusion processes not only smooth the noise but also blur the edges. An ideal diffusivity function allows more smoothing parallel to image edges and less smoothing perpendicular to these edges, which is actually anisotropic and nonlinear. For this reason, some classic milestones [33, 6, 48, 47] increase the nonlinear modeling ability by designing complex diffusivity functions $D = D(U)$ depending on the input $U$.

### 3.2 Rethinking Operator-level Boundary Awareness

Here we discuss the operator-level causes of potential CNN failures in boundary predictions. Given the input feature map $x$, the vanilla convolution calculates the output tensor $y$ through the inner product between the kernel $w$ and each local feature patch. Without loss of generality, we assume that the channel number of both input and output is 1. The vanilla convolution with kernel $w \in \mathcal{R}^{h \times w}$ over a input feature $x \in \mathcal{R}^{H \times W}$ could be written as: $y_p = \langle w, x_p \rangle$, where $x_p$ is the local patch centering at the position $p$. Intuitively, the vanilla convolution is just a local weighted average

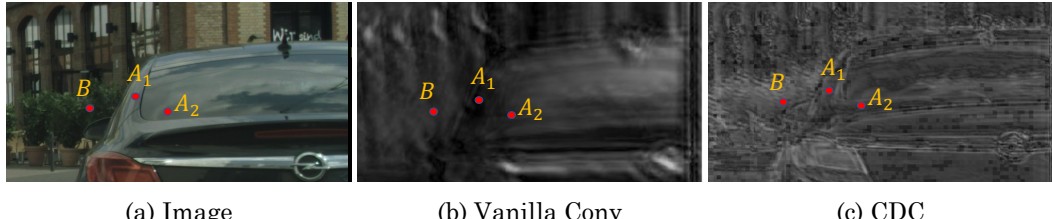

| (a) Image | (b) Vanilla Conv | (c) CDC |

Figure 2: Visualization of the feature maps from (b) vanilla convolution [24], (c) central difference convolution (CDC) [55]. We choose three points $A_1, A_2, B$ on the image, corresponding to *car-body*, *car-glass* and *plant*, respectively. In (b), due to the local smoothness of the vanilla-conv operator, we can see that there is no significant difference in the feature responses at the three coordinates in the feature map. According to (c), the CDC operator can hardly distinguish the true semantic boundary and the pseudo boundary caused by textures or noise, for example, we can see that the features captured by CDC have cluttered false boundary responses between these two coordinates $A_1$ (*car-body*) and $A_2$ (*car-glass*).

operation, which tends to smooth the local detailed information [55]. Especially when entries within the kernel are non-negative, the smoothness will be particularly significant as it forms a low-pass filter. This results in a poor capacity for modeling the semantic boundary information, see Figure 2b.

Recently, people try to combine the traditional gradient operator (edge-detector) with convolution operators to enhance its edge awareness [19, 53–56, 41]. Local binary convolution (LBC) [19] used a series of pre-defined binary filters to replace learnable kernels in convolution. Subsequently, a series of difference convolution operators represented by central difference convolution (CDC) [53–56] used learnable kernels to capture useful boundary information on the central difference map, that is, $y_p = \langle w, (x_p - x_{\text{center}}) \rangle$, where $x_{\text{center}}$ indicates the center entry of the current patch, and the rest symbols are defined in the same way as that of the vanilla convolution. It aggregates the center-oriented gradient within the local region instead of the feature values. Even though this kind of operator has better edge awareness, however, because the gradient operator cannot distinguish between real semantic boundary and pseudo boundary caused by textures or noise (see Figure 2c), it is difficult for us to directly apply them to tasks such as semantic segmentation.

## 4 Methodology

This section introduces the semantic diffusion network (SDN) for semantic segmentation tasks to compensate for the detail blur caused by the vanilla convolution operator. It is combined with CNN backbone when used, so as to strengthen semantic boundary information and filter out pseudo boundary information caused by texture or noise.

### 4.1 Formulation

Considering a CNN backbone model with $L$ stacked neural stages mapping an image $I$ to pixel-wise feature maps $X$. Let $X^\ell$ denote the feature map from the $\ell$-th stage. We define the semantic diffusion process over the feature map $X^\ell$ as:

$$\frac{\partial U_t}{\partial t} = Div\Big(g(|\nabla V|^2)\nabla U_t\Big), \tag{2}$$

where $\frac{\partial U}{\partial t}$ is the time derivative of the solution tensor, $Div$ is the divergence operator, $\nabla$ is the gradient operator. $V$ is the guidance feature map with the same resolution of $U$. The diffusivity term is a monotonically decreasing function of the square of the gradient $|\nabla V|^2$, for example, $g(|\nabla V|^2) = 1/\sqrt{1 + |\nabla V|^2/\lambda^2}$. Ideally, the diffusion close to the semantic boundary areas will be suppressed, while the diffusion far away from the semantic boundary will be accelerated. Therefore, $V$ should contain sufficient semantic information, so the gradient norm $|\nabla V|^2$ can be used as a reliable semantic boundary indicator to construct the diffusivity. The PDE of the non-linear diffusion process defined by Eq.(2) could be approximately solved by performing the finite difference update

rule for $T$ steps:

$$\widetilde{U}_{(m,n)}^{t+1} = \sum_{(i,j) \in \mathcal{N}_{(m,n)}} g(|V_{(i,j)} - V_{(m,n)}|^2) \cdot (U_{(i,j)}^t - U_{(m,n)}^t), \tag{3}$$

$$U_{(m,n)}^{t+1} = \alpha^t \cdot U_{(m,n)}^t + \beta^t \cdot \widetilde{U}_{(m,n)}^{t+1}, \tag{4}$$

where the step-index $t \in \{0, ..., T-1\}$, the tuple $(i,j)$ enumerates the index of every entry within the local neighbor region $\mathcal{N}_{(m,n)}$ of the position $(m,n)$, and the initial state is $U^0 = \boldsymbol{X}^\ell$. In fact, it is obvious that the diffusivity $g(|V_{(i,j)} - V_{(h,w)}|^2)$ calculates the similarity between semantic guidance features at $(i,j)$ and $(m,n)$. The solver builds a differentiable map from the input feature map $\boldsymbol{X}^\ell$ to the boundary-enhanced feature.

**Remark.** The finite difference based solver can hardly be directly integrated with deep segmentation models for several reasons: (1). the stability of the finite difference method depends on the setting of boundary conditions and the selection of parameters $(\alpha, \beta)$; (2). the solver requires multiple iterations, which will consume high computational complexity. (3). integrating the iterative solver with the deep networks would result in numerical instability to the training process, such as gradient explosion [39].

### 4.2 Semantic Diffusion Network

This subsection describes a novel learnable approach called **semantic diffusion network (SDN)** for approximating the diffusion process, which contains a parameterized semantic difference convolution operator followed by a feature fusion module and constructs a differentiable mapping from original backbone features to advanced boundary-aware features. As a learnable PDE approximate solver, the well-trained SDN could approximate the diffusion process in only one forward computation, reducing the computational complexity and improving the stability. Overall, we formulate the SDN as $\boldsymbol{F}^{\text{sdn}} = \textbf{SDN}(\boldsymbol{U}, \boldsymbol{V})$, where $\boldsymbol{U}$ is the input feature map to be processed and $\boldsymbol{V}$ is the semantic guidance map.

#### 4.2.1 Semantic Difference Convolution

Inspired by Eq.(3), we develop the semantic difference convolution (SDC) operator. We set the $\mathcal{N}_{m,n}$ as a $h \times w$ local region centered at position $(m,n)$. SDC takes the feature map $\boldsymbol{U}$ and the semantic guidance $\boldsymbol{V}$ as inputs and calculates the output feature map $\boldsymbol{Y}$. Specifically, each component of $\boldsymbol{y}$ indexed by $(m,n)$ is calculated via

$$\boldsymbol{Y}_{(m,n)} = \sum_{(i,j) \in \mathcal{N}_{(m,n)}} \boldsymbol{W}_{(i,j)} \cdot \boldsymbol{S}(\boldsymbol{V}_{(i,j)}, \boldsymbol{V}_{(m,n)}) \cdot (\boldsymbol{U}_{(i,j)} - \boldsymbol{U}_{(m,n)}), \tag{5}$$

where the tuple $(i,j)$ enumerates the index of every entry within the local neighbor region $\mathcal{N}_{m,n}$. In the right hand of Eq.(5), the first term is the learnable convolutional kernel $\boldsymbol{W}$, of which (height, width) $= (h, w)$. The kernel introduces a more powerful modeling capability than the naive non-linear diffusion process in Eq.(2). The second term $\boldsymbol{S}(\boldsymbol{V}_{(i,j)}, \boldsymbol{V}_{(m,n)})$, called the semantic similarity term, is a scalar measuring the similarity between semantic guidance features at $(i,j)$ and $(m,n)$. This term corresponds to the diffusivity $g(|V_{(i,j)} - V_{(h,w)}|^2)$ in Eq.(3). The third term $(\boldsymbol{U}_{(i,j)} - \boldsymbol{U}_{(m,n)})$, called the pixel difference term, calculates the feature difference between features at the position $(i,j)$ and $(m,n)$. Note that Eq.(5) can be implemented efficiently using CUDA. By default, the kernel size of $\boldsymbol{W}$ is set to $3 \times 3$.

Figure 3 provides the comparison between SDC with other well-known convolution operators. As defined in Eq.(5), the proposed SDC operator contains two sequential steps: (i) calculating the semantic similarity, and (ii) calculating the output feature. The first step consumes a complexity of $\mathcal{O}(HWhwC_f)$, and the second step consumes a complexity of $\mathcal{O}(HWhwC_iC_o)$. Thus, the overall complexity is Complexity $= \mathcal{O}(HWhw(C_iC_o + C_f))$, where $H, W$ are the height and width of the input feature, and $h, w$ are the height and width of the kernel tensor. $C_i, C_o$, and $C_f$ are the channel number of input, output, and semantic features, respectively.

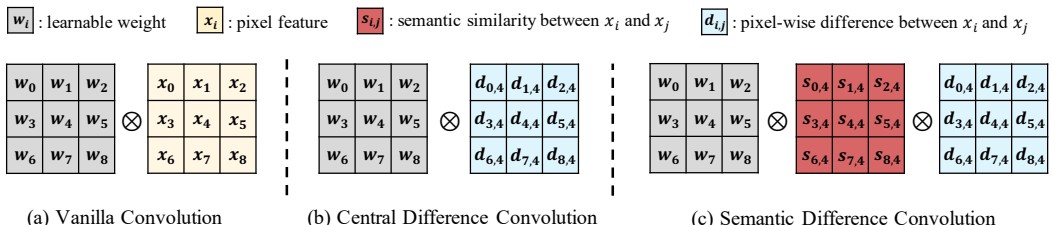

Figure 3: Comparisons between the vanilla convolution, the central difference convolution, and our semantic difference convolution.

#### 4.2.2 Feature Fusion

This step aims to fuse the semantic difference convolution output $\boldsymbol{Y}$ and the module's input feature map $\boldsymbol{U}$ into the final boundary-enhanced feature map $\boldsymbol{F}^{\text{sdn}}$. This module is actually corresponding to Eq.(4) in the classic finite differential solution. Here, we extend the simple weighted addition operation in Eq.(4) to a learnable and more flexible way, that is, $\boldsymbol{F}^{\text{sdn}} = \mathbf{Conv}\Big(\big[\boldsymbol{U}, \boldsymbol{Y}\big]\Big)$. It first concatenates the original input feature $\boldsymbol{U}$ with the semantic difference convolution output $\boldsymbol{Y}$ and then uses a simple but effective $1 \times 1$ convolution to achieve channel-wise feature fusion, resulting in the boundary-enhanced feature $\boldsymbol{F}^{\text{sdn}}$. Note that both $\boldsymbol{U}$ and $\boldsymbol{Y}$ should have the same shape. Otherwise, we need to perform the bilinear interpolation upsampling operation on the smaller one to align the shapes.

### 4.3 Segmentation Model with SDN

Here, we introduce how to integrate our SDN into the existing encoder-decoder-based segmentation models. For better compatibility, we treat SDN as a neck part between the encoder and decoder, without breaking the original design of the baseline network. According to the feature scales required by the decoder, existing segmentation models can be divided into two categories: (i) encoder with single-scale decoder (*i.e.,* ViT+Segmenter and ResNet+FCN); (ii) encoder with single-scale decoder, such as ResNet+SemanticFPN. Next, we discussed the two cases separately.

**SDN with single-scale decoder.** As shown in Figure 4(a), given an input image $\boldsymbol{I} \in \mathbb{R}^{3 \times H \times W}$, the encoder is first applied to it to extract the feature, resulting in $\boldsymbol{F} \in \mathbb{R}^{C' \times H' \times W'}$, where $H'$, $W'$ and $C'$ denote the height, width and channel numbers, respectively. We then incorporate our SDN on the encoder output feature $\boldsymbol{F}$ to enhance the boundary details, and the output of SDN $\boldsymbol{F}^{\text{sdn}}$ is sent to the decoder. Note that $\boldsymbol{F}^{\text{sdn}}$ has the same size of $\boldsymbol{F}$. As mentioned in Sec. 4.2.1, SDN requires an additional feature with rich semantics as guidance. To construct such a semantic feature $\boldsymbol{F}^s$ for SDN, we stack a lightweight layer $\Phi$ on the top of the encoder, where $\Phi$ is instantiated as a $3 \times 3$ convolution with stride 2. Thus, we have $\boldsymbol{F}^{\text{sdn}} = \mathbf{SDN}\big(\boldsymbol{F}, \boldsymbol{F}^s\big)$.

**SDN with multi-scale decoder.** As Figure 4(b), the input image $\boldsymbol{I} \in \mathbb{R}^{3 \times H \times W}$ is first passed through a backbone with totally $L$ scales (generally $L = 4$), resulting in multi-scale features $\{\boldsymbol{F}_i \in \mathbb{R}^{C_i \times H_i \times W_i}\}_{i=1}^{L}$. $H_i$, $W_i$ and $C_i$ denote the height, width, and channel numbers, respectively. We then incorporate the SDN after all the backbone stages to strengthen the boundaries comprehensively. In particular, for the $i$-th SDN, it takes $\boldsymbol{F}_i$ and $\boldsymbol{F}_i^s$ as the inputs, and generates the feature: $\boldsymbol{F}_i^{\text{sdn}} = \mathbf{SDN}\big(\boldsymbol{F}_i, \boldsymbol{F}_i^s\big), 1 \leq i \leq L$, where $\boldsymbol{F}_i^s$ denotes the semantic feature at scale $i$, which is defined as follows:

$$\boldsymbol{F}_i^s = \begin{cases} \boldsymbol{F}_{i+1}, & 1 \leq i < L \\ \Phi\big(\boldsymbol{F}_L\big), & i = L \end{cases} \tag{6}$$

Similarly, $\Phi$ is also implemented by a $3 \times 3$ convolution with stride 2. Note that the SDNs applied on different stages have the same architecture but do not share the same parameters. Finally, the boundary-enhanced features are fed into the decoder to generate the pixel-wise probability map.

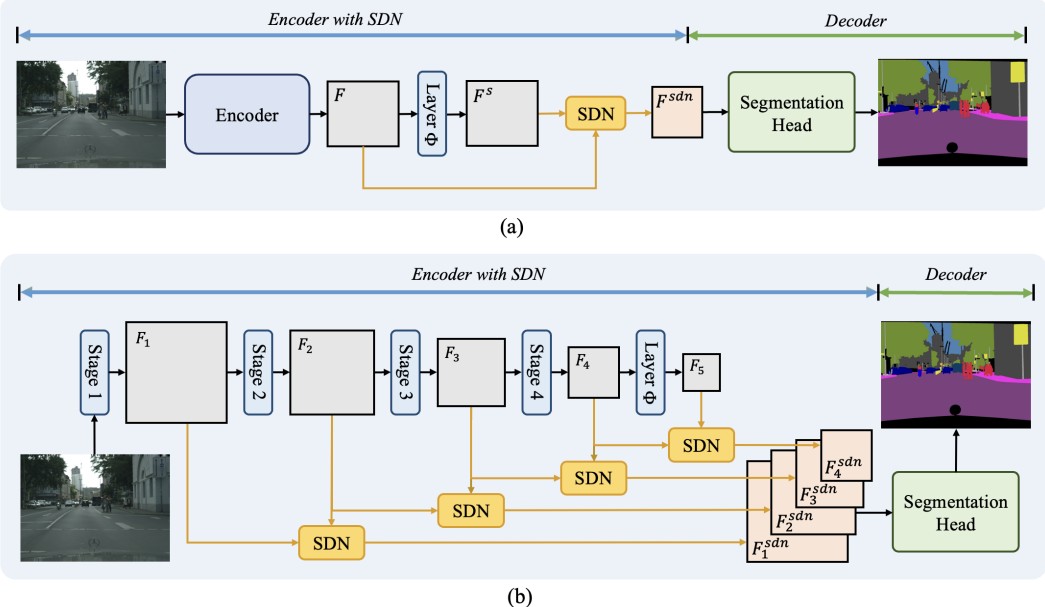

Figure 4: Illustration of how to combine the proposed semantic diffusion network (SDN) with the baseline segmentation model with single-scale decoder and multi-scale decoder in (a) and (b), respectively. The SDN is regarded as a neck part in order not to break the original encoder and decoder design of the baseline model.

## 5   Experiments

In this section, we first introduce the datasets and implementation details in Sec. 5.1. Then, we evaluate our approach over several baseline segmentation models on these public benchmarks in Sec. 5.2. Finally, in Sec. 5.3, we conduct ablation studies to analyze the effect of key designs and the compatibility of our approach.

### 5.1   Experimental Setup

**Datasets.** Experiments are conducted on two widely-used public benchmarks: *ADE20K* [62] is a very challenging benchmark including 150 categories, which is split into 20000 and 2000 images for training and validation. *Cityscapes* [9] is a real-world scene parsing benchmark, which contains over 5000 urban scene images with 19 classes. The number of images for training, validation, and testing are 2975, 500, and 1525, respectively.

**Baseline models.** To evaluate the effectiveness of our SDN, we integrate it into two typical CNN-based baseline methods (FCN [18] and SemanticFPN [21]) and a recent Transformer-based segmentation model (Segmenter [40]).

**Training.** All the experiments are implemented with PyTorch [1] and conducted on 4 NVIDIA V100 GPUs. For fair comparisons, we apply the same settings as the baseline methods. Specifically, for CNN baselines (*i.e.,* FCN and SemanticFPN), we use the SGD optimizer with momentum 0.9 and weight decay 0.0005. The learning rate is initialized at 0.01 and decays until 1e-4 by a polynomial strategy with power 0.9. For Transformer baselines (*i.e.,* Segmenter), AdamW is used as the optimizer without weight decay. The learning rate is initialized as 6e-5 and decays until 0 via a polynomial strategy with power 1.0. We train the models with 160k and 80k iterations for ADE20K and Cityscapes, respectively. For ADE20K, we train the models with 160k iterations, each of which involves 16 images. For Cityscapes, the training contains 80k iterations with a batch size of 8. Synchronized BN [32] is used to synchronize the mean and standard deviation of BN [38] across multiple GPUs. The backbone is initialized by the ImageNet-1K [36] and ImageNet-22K [10] pre-trained weights for ResNet-50 [14] and ViT-B [11], respectively.

Table 1: Experimental results on ADE20K-*val* and Cityscapes-*val*. "s.s." and "m.s." denote the single-scale and multi-scale mIoU, respectively.

| Method | Encoder | ADE20K | | Cityscapes | |
| | | mIoU (s.s.) | mIoU (m.s.) | mIoU (s.s.) | mIoU (m.s.) |
|---|---|---|---|---|---|
| FCN [18] | ResNet-50 | 36.10 | 38.08 | 72.64 | 73.32 |
| **FCN+Ours** | ResNet-50 | **38.12 (+2.02)** | **39.36 (+1.28)** | **74.75 (+2.11)** | **75.79 (+2.47)** |
| SemanticFPN [21] | ResNet-50 | 37.49 | 39.09 | 74.10 | 75.98 |
| **SemanticFPN+Ours** | ResNet-50 | **38.79 (+1.30)** | **40.27 (+1.18)** | **75.97 (+1.87)** | **77.31 (+1.33)** |
| Segmenter [40] | ViT-B | 48.48 | 50.00 | 77.97 | 80.07 |
| **Segmenter+Ours** | ViT-B | **51.05 (+2.57)** | **52.18 (+2.18)** | **79.42 (+1.45)** | **81.38 (+1.31)** |

Table 2: Boundary F-score comparisons on Cityscapes-*val*.

| Method | Encoder | with SegFix [58] | with SDN (ours) | F-score | |
| | | | | 1px | 3px |
|---|---|---|---|---|---|
| OCRNet[57] | HRNet-48[42] | ✗ | ✗ | 65.2 | 76.7 |
| | | ✓ | ✗ | 66.7 (+1.5) | 77.1 (+0.4) |
| | | ✗ | ✓ | **69.5 (+4.3)** | **78.2 (+1.5)** |

**Evaluation.** We adopt the widely-used mean intersection of union (mIoU) to measure the overall segmentation performance. In order to measure the boundary quality of the prediction, we compute F-score only on the 1px and 3px boundary region, which denotes the 1 pixel and 3 pixels width area along the ground-truth boundary. For the multi-scale inference, we follow the previous works [57, 8] to perform horizontal flip and average the predictions at multiple scales [0.5, 0.75, 1.0, 1.25, 1.5, 1.75, 2.0].

### 5.2 Main Results

To verify the effectiveness and generalization of our approach, we apply the proposed SDN to several baseline segmentation models on various datasets. Table 1 reports the quantitative comparisons.

**ADE20K.** Table 1 shows that our approach brings significant mIoU improvements for all the baselines on the challenging ADE20K dataset, *i.e.*, improves FCN by +2.02% mIoU (s.s.), improves SemanticFPN by +1.30% mIoU (s.s.), and improves Segmenter by +2.57% mIoU (s.s.). The improvement could be attributed to the fact that explicit boundary information is particularly useful for ADE20K images with complex scenarios. With the HRNet-48, ResNet-101, and Swin-B backbone, we contribute +2.6%, +1.7%, and +1.1% mIoU gain over OCRNet and MaskFormers, respectively.

**Cityscapes.** Table 1 shows that our approach brings nontrivial mIoU improvements for all the baselines, *i.e.*, improves FCN by +2.47% mIoU (m.s.), improves SemanticFPN by +1.87% mIoU (s.s.), and improves Segmenter by +1.45% mIoU (s.s.). Additionally, with the OCRNet(HRNet-w48) [57] baseline, SDN brings +0.9% mIoU improvement to the baseline OCRNet (81.9% to 82.8% mIoU) and outperforms the SegFix model by +0.3% mIoU (the OCRNet's results are not listed in Table 1). Moreover, our approach (84.7% mIoU) outperforms the OCRNet+SegFix (84.5% mIoU) by +0.2% on Cityscapes-*test*, where the detailed results can be found in the web-link. We also compare the boundary-promoting ability of SDN and the classic SegFix [58] model in Table 2, where our method brings significant improvement, *i.e.*, +4.3 and +1.5 for OCRNet in terms of the 1px and 3px F-score. Figure 5 provides some typical visualizations to show the boundary improvement effect of our method.

### 5.3 Further Study

**Compare SDC with other operators.** For direct comparisons with other related operators, we replace the semantic difference convolution (SDC) in SDN with other two related operators, vanilla convolution [24] and CDC [55]. Table 3(a) compares their boundary performance on Cityscapes

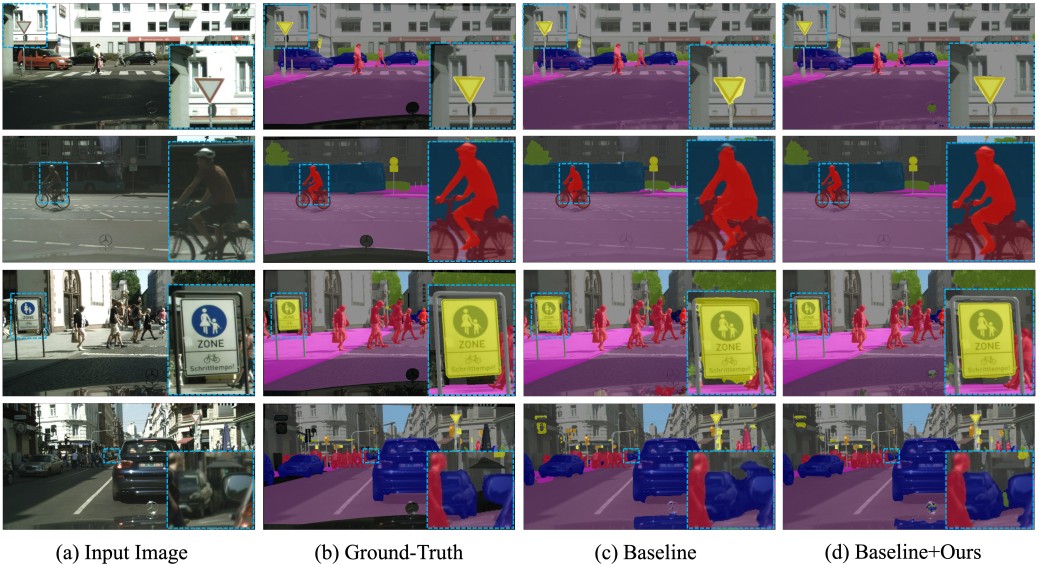

| (a) Input Image | (b) Ground-Truth | (c) Baseline | (d) Baseline+Ours |

Figure 5: Visualization comparisons of the segmentation results on Cityscapes-*val*, where the baseline is OCRNet.

with OCRNet [57] as the baseline model. It can be found that SDC outperforms its vanilla convolution counterpart by +4.3 F-score, and surpasses CDC by a large margin, which demonstrates the irreplaceability of SDC.

**Effect of kernel settings of SDC.** We conduct experiments under different choices of the kernel size and dilation rate of SDC in Table 3(b). It can be found that: (1). Increasing the size of the convolution kernel in SDC does not bring significant changes. (2). Excessive dilation rate leads to negative effects. In fact, large kernels and dilated convolution are mainly used for enlarging the receptive fields. While, the goal of our SDC is to infer the inter-class boundary cue in a local region, which may not require more contexts brought by large kernels. Similarly, a large dilation rate may lead to the lack of local information, where, however, the local information is critical for local boundary exploration.

**Compatibility with other boundary promoting methods.** Our approach can be incorporated with other boundary improvement methods, for example, using our SDN and a post-processing boundary refinement module simultaneously in a network. Table 3(c) presents the good compatibility of our approach with three well-known methods on Cityscapes-*val*. Specifically, the additional DenseCRF [23], SegFix [58] and InverseForm [3] further bring +0.2, +0.6 and +1.0 boundary F-score gains, respectively. It is worth noting that, our method can directly achieve +4.4 boundary F-score improvement over the baseline without any other methods. Benefiting from the intrinsic boundary exploration, segmentation networks with our SDN can relieve the dependence on other boundary improvement techniques.

Table 3: Further studies on (a) comparison with other operators, where V-Conv denotes the vanilla convolution, (b) the effect of the kernel size and dilation rate settings of the SDC operator in the SDN module, and (c) compatibility with other boundary-promoting methods in semantic segmentation.

(a)

| Operator | F-score |
| --- | --- |
| Baseline | 65.2 |
| V-Conv [24] | 65.2 |
| CDC [53] | 60.1 |
| SDC (ours) | 69.5 |

(b)

| Kernel-size | Dilation-rate | F-score |
| --- | --- | --- |
| $3 \times 3$ | 1 | 69.5 |
| $5 \times 5$ | 1 | 69.6 |
| $3 \times 3$ | 3 | 69.1 |
| $3 \times 3$ | 5 | 68.8 |

(c)

| Model | F-score |
| --- | --- |
| Baseline | 65.2 |
| + Ours | 69.5 |
| + Ours + CRF [23] | 69.7 |
| + Ours + SegFix [58] | 70.1 |
| + Ours + IF [3] | 70.5 |

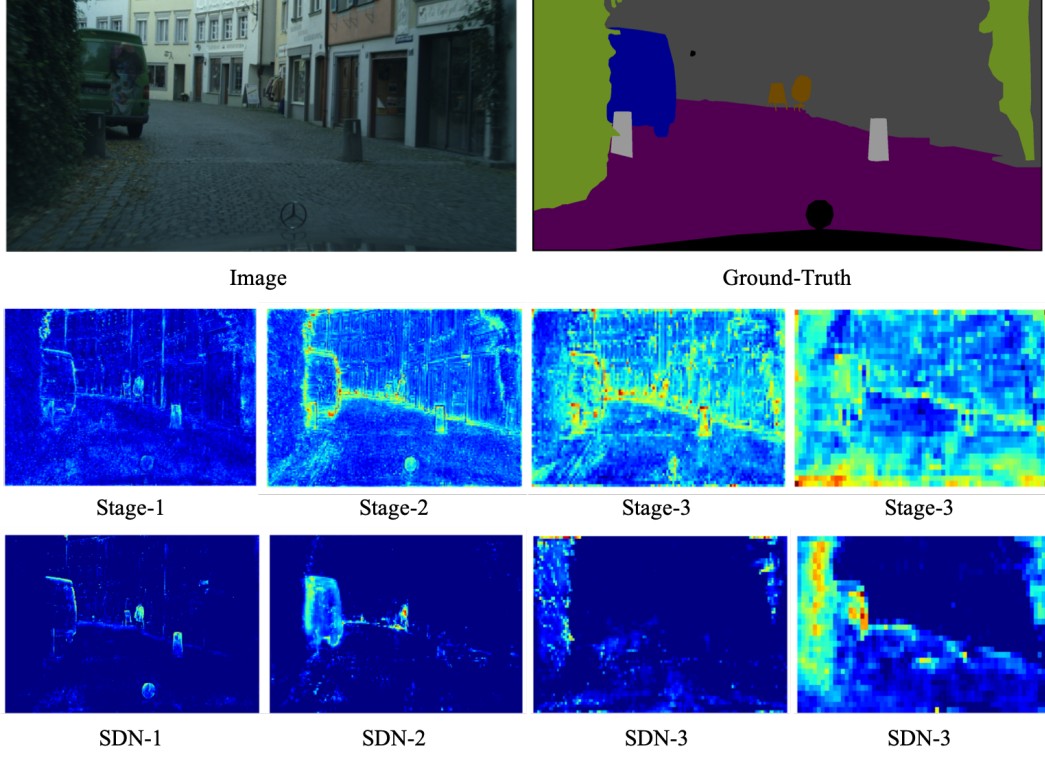

Figure 6: Visualization of the feature mapping in SDN. The second row shows the output feature of each encoder stage, which is also the input feature of SDNs. The output features of SDN according to each stage are shown in the third row. Obviously, our SDN enhances inter-class boundaries and suppresses the intra-class pseudo boundaries. The example is selected from Cityscapes.

## 6    Conclusion

In this paper, we have summarized the internal reasons why the deep semantic segmentation model is difficult to produce accurate boundary prediction, that is, the commonly used convolutional operators tend to smooth and blur local detail cues, making it difficult for deep models to capture precise boundary information. To solve this problem, this paper introduces an operator-level approach to enhance semantic boundary awareness, so as to improve the prediction of the deep semantic segmentation model. Specifically, we formulate the boundary feature enhancement process as an anisotropic diffusion process and propose the novel semantic diffusion network (SDN) to approximate the anisotropic diffusion process. The proposed SDN is an efficient, stable, and flexible module that can be plugged into existing encoder-decoder segmentation models. Extensive experiments show that our approach can achieve consistent improvement over various baseline semantic segmentation models and significantly improve the boundary quality of segmentation results.

## 7    Limitations and Social Impact.

We believe that SDN is a general approach that can benefit deep learning models in various popular visual tasks. However, this paper only verifies the effectiveness of SDN on a single task (semantic segmentation). In the recent future, we will explore the application of SDN in other 2D/3D tasks, such as instance segmentation, panoptic segmentation, flow estimation, image de-noising, image super-resolution, and so on. SDN has potential value in important applications such as autonomous driving and medical image analysis. But it can also deploy inhumane surveillance. The potential negative effects can be avoided by implementing strict and secure regulations.

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
