# OpenReview forum: "Semantic Diffusion Network for Semantic Segmentation"
_NeurIPS.cc/2022/Conference — NeurIPS 2022 Accept_

### Official Review · Reviewer_U8Zf · 2022-07-07

**Rating:** 7
**Confidence:** 4
**Soundness:** 4 excellent
**Presentation:** 3 good
**Contribution:** 4 excellent

**Summary:**

Overall, this is a good work.

In this paper, a new semantic segmentation pipeline is presented, based on proposed semantic difference convolution. The semantic difference convolution incorporates higher-level semantics into the lower-level convolution and helps learn better features which are sensitive to semantic boundaries. The proposed method is easy to implement on existing feed-forward segmentation baselines. According to a set of experiments on previous segmentation models including DeepLabv3, OCRNet and MaskFormer on 4 public datasets, the effectiveness of the proposed method is clearly demonstrated. Besides, this paper also provides analysis on the computational complexity and various comparisons regarding the configurations, as well as many ablations to show the SOTA results from the proposed method.

This paper is also well-organized, that I can follow their draft easily. Even though there are a few comments from my side, that I believe it could be improved, this is a good work for NeurIPS.

After reading all the comments and authors' feedback, I vote for accept (7) on this submission.

**Questions:**

1. This paper applies DeepLabv3, why not DeepLabv3+?
2. In Table 5 (b), why is the baseline performance only 65.1? What is the baseline here? More details are needed. Besides, why not apply a stronger baseline in Table 5?

**Limitations:**

No. This work is a fundamental research in computer vision.

**Strengths And Weaknesses:**

Strengths:
+ This paper is well organized. It is overall easy to follow.
+ This paper proposes a new convolution operation, which is simple to implement. Based on the proposed semantic difference convolutions (SDC) and SDM, more favorable segmentation results can be achieved. To show the effectiveness of the proposed method, the authors show different network backbone as well as segmentation models.
+ The proposed SDC achieves more improvements compared with previous boundary-aware methods. Further, the SDC is compatible with other techniques, such as DenseCRF, SegFix.
+ From the analysis and empirical evaluation, the proposed SDC will not increase the computational cost significantly.

Weakness:
- In section 3.2, it needs to discuss more details of f_{pi}. What is f_{pi}? Even though readers can understand when they see section 4.2 and Figure 4, it is a little confused in section 3.2. More explanation on the semantic information f_{pi} will be helpful.
- Lack of comparison with more boundary-aware methods, such as Gated-SCNN.
- In this work, DeepLabv3, OCRNet and MaskFormer are applied as baseline models. Why not apply the proposed method on the current state-of-the-art model to achieve new SOTA result, for example SegFormer?

---

> ### Author Response · Authors · 2022-08-02
> **Response to ''Weakness'' comments**
>
> Sincerely thanks for your appreciation of our work. Hoping our response will address your concerns.
>
> ---
>
> **1: Further clarification of the semantic feature $f_{pi}$.**
>
> Thanks for your advice. \
> We will add the necessary description in the revised version according to your suggestion. An academic consensus is that as the number of layers increases, features will contain more high-level semantic information (category-level, object-level). In principle, SDC/SDM used the deeper features of the deep model to refine the lower-level features; for example, in our SDC/SDM, the feature map of stage $i+1$ was used as the semantic feature (semantic guidance) to process the features of stage $i$.
>
>
> ---
>
> **2: Further comparison with more boundary-aware methods, such as Gated-SCNN.**
>
> Thanks for your valuable comments! \
> Here, we report the comparison between our approach and the Gated-SCNN under the same baseline (DeepLab v3+ with WideResNet as the backbone). The training protocol is consistent with that in Gated-SCNN.
> Our approach outperforms the Gated-SCNN by +0.8% mIoU.
>
>
> | Model    |  mIoU (m.s.)     |
> | :---- | :----: |
> Gated-SCNN    |   80.8  |
> Ours |  81.6 (+0.8)  |
>
>
>
> ---
>
> **3: Further comparison with SOTA, e.g., SegFormer.**
>
> Thanks for your advice and we will add this to the revision. \
> Here we choose the SegFormer as the baseline to evaluate our approach’s performance on ADE20K. We found that our SDM can improve the performance of Segformer by +0.9$\sim$+1.4% mIoU when combining backbone models with different scales.
>
> | Backbone |  SegFormer   |  SegFormer + Ours
> | :----: | :----: | :----: |
> | MiT-B0      | 37.4   |   38.8  (+1.4)   |
> | MiT-B4    | 51.1   |   52.1  (+1.0)  |
> | MiT-B5      | 51.8   |   52.7 (+0.9)   |

---

> ### Author Response · Authors · 2022-08-02
> **Response to ''Questions'' comments**
>
> Sincerely thanks for your appreciation of our work. Hoping our response will address your concerns.
>
> ---
>
> **1: Further comparison with DeepLab V3+.**
>
> Thanks for your advice. We will add it in the revised paper. \
> Here, we add a comparative experiment with DeepLab V3+ (ResNet-101) in the table below, hoping to solve your concerns. Experiments show that our approach can bring a +1.2% mIoU performance improvement when using DeepLab V3+ (ResNet-101) as the baseline.
>
>
> | Model     | mIoU (m.s.) |
> | :---- | :----: |
> | DeepLab V3+ | 80.5 |
> | DeepLab V3+ + Ours | 81.7 (+1.2)  |
>
> ---
>
> **2: About the baseline performance in Table 5(b).**
>
> Thank you for your careful review. \
> Table 5(b) reports the Boundary F-score of each method within a width of 1-px around the boundary. In fact, all current segmentation methods perform poorly on this metric. In contrast, the Boundary F-scores of the various methods improved considerably over a wider area (e.g., 3-px, 6-px, 9-px). In Table 3, we also compared the boundary F-score of two baselines under the 3-px mode and our method can still significantly improve the baseline even under the 3-px mode.

---

> ### Author Response · Authors · 2022-08-07
> **Looking forward to hearing from Reviewer U8Zf!**
>
> Dear reviewer U8Zf:
>
> Hello!
>
> First of all, we sincerely thank you for your appreciation of our work! We are honored by this!
>
> In our response comments, we have carried out explanations and experiments for your suggestions and concerns, including the further discussion about the semantic feature $f$, the further comparison between ours and other works (Gated-SCNN, DeepLab V3+, and SegFormer), further explanation of table 5(b).
>
> We sincerely look forward to hearing from you. If there is any question or concern, please feel free to let us know. We will try our best to address your concerns. At the same time, if you are satisfied with our work and response, we hope to obtain your further recognition and a higher rating score from you. We also hope that this article can be honored to receive the attention and recognition of the entire academic community!
>
> Best regards.
> The authors of “Semantic Difference Convolution for Semantic Segmentation”.

---

> > ### Comment · Reviewer_U8Zf · 2022-08-08
> > **Thanks for resolving my concerns**
> >
> > Thanks a lot for the efforts in addressing all the concerns raised by reviewers!
> >
> > The authors show more experimental results on SOTA models, such as DeepLab-v3+ and SegFormer.
> >
> > I do not have more concerns on the novelty of this work, even though the contribution is incremental based on DC. However, it introduces semantics into the operation, which leads to improved segmentation performance and supposed to accept by NeurIPS.
> >
> > The authors should improve the writing for publication. By reading all the comments and feedback, I insist on my suggestion and believe this work should be accepted.

---

> > > ### Author Response · Authors · 2022-08-09
> > > **Thanks for Reviewer U8Zf**
> > >
> > > Dear Reviewer:
> > >
> > > &nbsp;
> > >
> > > Sincerely thanks for all your efforts and suggestions! They are important for us to further improve this paper!
> > >
> > > It's a great honor to get your recognition! Thanks for raising the score. We will try our best to improve the final version of the paper.
> > >
> > > &nbsp;
> > >
> > > Best regards. \
> > > The authors of “Semantic Difference Convolution for Semantic Segmentation”.

---

### Official Review · Reviewer_GWNK · 2022-07-11

**Rating:** 6
**Confidence:** 4
**Soundness:** 3 good
**Presentation:** 3 good
**Contribution:** 3 good

**Summary:**

This paper studies the problem of how to improve the semantic segmentation results for existing architectures. The authors propose a semantic difference guided convolution, so-called SDC, to deliberately enhance the representations in the semantic boundary area. Specifically, it enhances current layer’s kernel with the the next layer’s feature differences, instead of using current layer’s differences like previous methods, which means that the next layer’s features could serve as a semantic guidance to the current layer. The proposed module is claimed to be flexible to co-operate with any ‘encoder-decoder’ architectures. And the effectiveness is verified through extensive experiments.

**Questions:**

Please refer to the Strengths and weakness.

**Limitations:**

The authors have pointed out the limitations of their work.

Since it is just a basic method to deal with 2D semantic segmentation, it has nothing to do with the potential negative societal impact.


**Strengths And Weaknesses:**

** Strength **

+: The paper is well written and easy to follow.

+: The insight of enhancing the semantic boundary through next stage’s differences is interesting and has somewhat novelty for the semantic segmentation. Can this method generalize to other dense estimation task?

+: The authors fully demonstrate the effectiveness of the proposed method on various datasets.

** Weakness **

-: The wording of “intra-class boundary“ is not precise enough. According to my understanding, the semantic difference is provided by the next layer’s feature, not the output layer. And only the output layer corresponds to the class-level information exactly. Especially in the early layer, two approaching pixels with different features do not imply they belong to different classes.

-: There seems to be no normalization to the SD(*). Would this induces unstable training if the SD(*) generates a quite large number?

-: In table 5(b), it would be more convincing if the author organizes the comparisons as “Baseline, Baseline+Ours, Baseline+DenseCRF, Baseline+DenseCRF+Ours, Baseline+SegFix, Baseline+SegFix+Ours, Baseline+InverseForm, Baseline+InverseForm+Ours”.

---

> ### Author Response · Authors · 2022-08-02
> **Response to ''Questions (Weakness)'' comments**
>
>
> Thank you sincerely for your appreciation and constructive comments on this paper.
> We will do our best to address your concerns, and revise the paper according to your suggestion.
>
> ---
>
> **1: Can this method generalize to other dense estimation tasks?**
>
> Actually, the proposed SDC and SDM can also be applied to various vision tasks, such as instance segmentation, which are served as our future work.
>
> ---
>
> **2: Concerns about the wording of “intra-class boundary.”**
>
> This is a good question! \
> We agree with you that only the output layer contains exact category-level information.
> The feature before projecting to the class number dimension also contains the implicit category information.
> In the beginning, we tried to add an auxiliary head supervised by cross-entropy loss between ground-truth and prediction for the output feature of each stage of the backbone and use the output score of auxiliary heads to compute the semantic difference term.
> We found that the performance is similar to using the feature directly while introducing too many parameters attributed to the additional auxiliary heads especially when the dataset has a large number of classes.  Besides, as the features of shallow layers (stage 1 and 2 of the backbone) lack enough semantic information, the segmentation performance of their corresponding auxiliary heads is poor.
> Therefore, we choose the current simple but effective implementation.
>
> ---
>
> **3: Concerns about the potentially unstable training caused by the unnormalized SD term.**
>
> Great question! \
> Actually, we have not found the unstable training phenomenon that you are concerned about.
> The reason may be that even though the semantic feature fed into SD is not normalized, the BN layer, which is widely used in the whole network, avoids the input features in SD from producing abnormal changes. \
> We have also compared some different measures of semantic difference term in Table 4 of the supplementary material. Specifically, we replace the Euclidean distance with cosine distance with a limited range of $[0, 1]$. And we found that there was no significant difference. For clarity, we list the main results in the table below.
>
> | measure of semantic difference term    |  mIoU  |
> | :---- | :----: |
> |  Euclidean Distance  |  82.9   |
> |  Cosine Distance  | 82.8 |
>
> ---
>
> **4: Further experimental results in Table 5(b).**
>
> Thank you for your good suggestion!  In the following table, we reorganized Table 5(b) as you recommended.
>
>  | Model   | Boundary F-score
> | :---- | :----: |
> | Baseline      |  65.1
> | Baseline+Ours      |  69.5
> | Baseline+DenseCRF      |  67.2
> | Baseline+DenseCRF+Ours      |  69.7
> | Baseline+SegFix      |  68.8
> | Baseline+SegFix+Ours      |   70.1
> | Baseline+InverseForm      |  69.1
> | Baseline+InverseForm+Ours      |  70.5

---

> > ### Comment · Reviewer_GWNK · 2022-08-09
> > **Response to author rebuttal**
> >
> > Thanks for the detailed rebuttal. I addresses my concern. So I keep my rating as weak accept.

---

> > > ### Author Response · Authors · 2022-08-09
> > > **Thanks for Reviewer GWNK**
> > >
> > > Dear Reviewer:
> > >
> > > Thanks a lot for your time and efforts in reviewing our paper! \
> > > It's really good news that our reply has addressed your concerns. If you have any other questions, please feel free to contact us, we will try our best to address them in time! \
> > > Sincerely thanks for your appreciation again! We are honored by this!
> > >
> > > Best regards! \
> > > The authors of “Semantic Difference Convolution for Semantic Segmentation”.

---

> ### Author Response · Authors · 2022-08-07
> **Hope for hearing from Reviewer GWNK!**
>
> Dear reviewer GWNK:
>
> Hello!
>
> First of all, we would like to express our heartfelt thanks to you for your appreciation of this work and your constructive suggestions!
>
> In our response comments, we have carried out explanation and experiments based on your suggestions and concerns, including the further discussion about the wording of “intra-class boundary”, explanation (including the analysis and experiments) of the normalization in the SD-term, and further comparisons in Table 5(b).
>
> We sincerely look forward to hearing from you. If there is any question or concern, please feel free to let us know. We will try our best to address your concerns. At the same time, if you are satisfied with our work and response, we hope to obtain your further recognition and a higher rating score from you.
>
> Best regards.
> The authors of “Semantic Difference Convolution for Semantic Segmentation”.

---

### Official Review · Reviewer_XkWA · 2022-07-11

**Rating:** 7
**Confidence:** 4
**Soundness:** 3 good
**Presentation:** 3 good
**Contribution:** 3 good

**Summary:**

This paper proposes an efficient boundary-aware convolution operator to boost the boundary modeling capacity for semantic segmentation. The proposed operator is sensitive to the inter-class boundary while ignoring the noisy intra-class pseudo-boundaries. Based on the proposed operator, a lightweight module is designed to enhance boundary-related information, which is flexible to be plugged into any existing encoder-decoder segmentation model. Experimental results validate that  the proposed approach can achieve consistent improvements for boundary regions over several baselines.



**Questions:**

-	The further analyses in Table 5a are appreciated, it will be more convincing if some details are given. How to calculate $\rou$ in this table? How many percents of more parameters are introduced by the proposed approach for each convolution layer? Does CDC also introduce more parameters to the network?
-	About the pure improvement of the semantic difference term: If we replace SDC with CDC in the proposed approach, how well does it perform?
-	Does the benefit of the semantic difference term for intra-class boundaries rely on the scale of convolution kernels and the quality of the semantic feature?


**Limitations:**

The authors adequately addressed their work's potential negative social impact.

**Strengths And Weaknesses:**

Strength:
-	This paper proposes an efficient boundary-aware convolution operator to boost the boundary modeling capacity for semantic segmentation.
-	The proposed approach shows promising performance for inter-class boundaries.
-	Comprehensive ablative studies are conducted to show the effectiveness of the proposed approach.

Weakness:
-	The proposed approach may introduce new parameters to the network.
-	The proposed Semantic Difference Convolution (SDC) is similar to Central Difference Convolution (CDC).
-	The effectiveness of the semantic difference term needs more clarification.

---

> ### Author Response · Authors · 2022-08-02
> **Response to ''Weakness'' comments**
>
> Thank you for your appreciation!  We will try our best to address your concerns!
>
> ---
>
> **1: About the new parameters introduced by our approach.**
>
> A good question! \
> Actually, our SDM is a quite efficient and lightweight module.
> As shown in Table 5(a), our SDM can bring +1.8% mIoU and +4.4% boundary F-score (1-px) for the well-known baseline segmentation model, OCRNet (HRNet-48), on Cityscapes-val dataset, while only introduces 0.1G FLOPs and 3.1M parameters. \
> Considering the significant performance gain, we believe that such few additional computational cost is well worth.
>
> ---
>
> **2: Comparisons between SDC and CDC.**
>
> This is a valuable question!  We will add the full comparisons between them into the revised version. \
> Next, we analyze the differences between SDC and CDC from the following four aspects:
> + **Motivation:**
>     - CDC is proposed for detecting the edge information effectively, which is sensitive to all edges including both inter-class boundaries and inner-class boundaries, thus can not be applied for semantic segmentation tasks.
>     - While, our SDC is specially designed for semantic segmentation tasks, which can enhance only inter-class boundaries and suppress inner-class boundaries.
> + **Formulation:**
>     - As shown in Eq.(2) and Eq.(3), compared with CDC, our SDC has an additional term, i.e., semantic difference term, which is simple but the most critical point to filter out the perception of pseudo-inner-class boundaries.
> + **Application:**  According to their properties, SDC and CDC have completely different application scenarios.
>     - With the ability to perceive only inter-class boundaries, our SDC is more suitable for various visual tasks to distinguish objects with different classes, such as semantic segmentation.
>     - By contrast, CDC is better suited for detecting class-agnostic edges and has been widely used for edge detection and face-spoof detection tasks.
> + **Theoretical analysis:**
>     - Inspired by image filtering theory, we modeled feature fusion and boundary enhancement of SDC/ SDM as an anisotropic diffusion model in classical physics. Further, we give a theoretical explanation of SDC/SDM, that is, SDC/SDM acts as a stable and efficient PDE solver. The details are updated in the revised supplementary material.
>     - In contrast, CDC is derived from the gradient operator/Laplacian operator in traditional image processing.
>
> ---
>
> **3: Further clarification of the effectiveness of the semantic difference term (SD-term).**
>
> Thanks again for your advice! \
> We will give more explanations in terms of the following three aspects:
> + **Empirical analysis:** Compared with features of shallow-layer, deep features generally contain higher-level semantic information, which is closer to category-level and object-level. The SD-term in SDC uses the semantic feature to selectively filter out false boundaries and enhance real boundary features. For two different pixels inside the same object, their high-level semantic features tend to be similar. In this case, the SD-value will be small, and the SD-value will suppress the Pixel-wise Difference item, which plays the role of suppressing the inner-object variation. In the face of two pixels located in objects of different categories, their high-level features are often different, and the SD-value will be very large. In this case, even if the Pixel-wise difference of the low level may be small, the large SD-value will highlight and strengthen it, which plays a role in strengthening the boundary of the real category.
> + **Ablation experiments:** We also conducted ablation experiments to verify the effectiveness of SDC (SD-term). For details, see Table 4 (a). After we replace the SDC operator in the SDM module with CDC(CDC is without the SD-term), the final performance will plummet from 82.9% mIoU to 80.4% mIoU!
> + **Theoretical explanation:** Inspired by the image filtering theory, our SDC/SDM actually models the anisotropic diffusion process in classical physics. Further, we give a theoretical explanation of SDC/SDM, that is, SDC/SDM acts as a stable and efficient PDE solver, and SD-term acts as an input in the diffusion coefficient function. The details were updated in the revised supplementary material.

---

> > ### Comment · Reviewer_XkWA · 2022-08-09
> > **Response to the rebuttal**
> >
> > Thanks for the detailed rebuttal. It has addressed all my concerns. Therefore  I keep my rating as "accept".

---

> > > ### Author Response · Authors · 2022-08-09
> > > **Thanks for Reviewer XkWA**
> > >
> > > Dear reviewer:
> > >
> > > Sincerely thanks for your appreciation again! We are honored by this! \
> > > It's really good news that our reply has addressed your concerns. Thanks for your time and valuable suggestions! They are critical to for us further improve this paper.
> > >
> > > Best regards!  \
> > > The authors of “Semantic Difference Convolution for Semantic Segmentation”.

---

> ### Author Response · Authors · 2022-08-02
> **Response to ''Questions'' comments**
>
> Thanks for your advice!  We are pleased to address all your concerns.
>
> ---
>
> **1: Further analyses for Table 5(a).**
>
> Thanks for your valuable advice!  We will explain it more clearly in the final version.
> + **The calculation of $\rho$:**  As the caption of Table 5, the $\rho$ in the 4-th column of Table 5(a) denotes the proportion of the computation cost introduced by our approach in the whole network, which is calculated by ("Baseline+Ours" - "Baseline") / "Baseline+Ours". Taking the 4.22% in the first row of Table 5(a) as an example, it is obtained by $\frac{73.5 - 70.4}{73.5}$.
> + **Percents parameters introduced for each convolution layer:**   In fact, our SDC does not introduce additional parameters, and has the same parameters as vanilla convolution and CDC, as the semantic distance (SD) term in Eq.(3) is parameter-free. In each SDM, the parameters come from the two $1\times1$ convolution $\phi$ and $\psi$ in Eq.(6) and Eq.(8), which aim to project the semantic feature and fuse the input feature with the boundary-aware feature, respectively.
> + **Does CDC also introduce more parameters to the network?**   As mentioned above, both CDC and our SDC do not bring more parameters and have the same parameters as vanilla convolution. Thus, if we replace the SDC in SDM with CDC, it will introduce additional parameters as well.
>
> ---
>
> **2: About the pure improvement of the semantic difference term, and the performance of replacing SDC with CDC.**
>
> Great question! This is no doubt a critical ablation study. \
> Actually, we have conducted such experiments to verify the effectiveness of the proposed semantic difference term by replacing the SDC in SDM with CDC and vanilla convolution. \
> Please refer to Table 4 (a) in the main paper for details. After replacing SDC in SDM with CDC (CDC has no semantic difference term), the performance will drop abruptly (SDC: 82.9% mIoU, CDC: 80.4% mIoU, Vanilla-Conv: 81.3% mIoU, baseline: 81.1% mIoU)!
> The reason is that CDC tends to enhance all the edges including the pseudo boundaries (such as inner-object texture), which disrupts the semantic segmentation task. By contrast, benefiting from the semantic difference term in our SDC, only the real inter-class boundaries are strengthened, which is compatible with the goal of the semantic segmentation task.
>
> ---
>
> **3: Does the benefit of the semantic difference term for intra-class boundaries rely on the scale of convolution kernels and the quality of the semantic feature?**
>
> Sincerely thank you for your interesting question! We hope to dispel your concerns through our answers!
>
> ***(1) Effect of the kernel size and dilation rate of SDC:*** \
> We conduct experiments under different choices of the kernel size and dilation rate of SDC in the table below. It can be found that:
> + Increasing the size of the convolution kernel in SDC does not bring significant improvement (82.9 $\to$ 83.0).
> + Excessive dilation rate leads to negative effects (82.9 $\to$ 82.8 $\to$ 82.5).
>
> In fact, large kernels and dilated convolution are mainly used for enlarging the receptive fields.
> While, the goal of our SDC is to infer the inter-class boundary cue in a local region, which may not require more contexts brought by large kernels. Similarly, a large dilation rate may lead to the lack of local information, which is critical for local boundary exploration.
>
> | kernel size    |  dilation rate  |  mIoU  |
> | :----: | :----: | :----: |
> |  $3\times3$  |       1        | 82.9    |
> |  $5\times5$  |       1        | 83.0    |
> |  $3\times3$  |       3        | 82.8    |
> |  $3\times3$  |       5        | 82.5    |
>
> ***(2) Effect of the quality of semantic feature:*** \
> Definitely, we found that the quality of semantic features would significantly affect the benefits brought by semantic difference term.
> As we know, high-level features tend to contain more semantic information, so we design SDM as a neck-part to refine the features of the lower-level stage by using the features from the higher-level stage. \
> In Table 5b, we have compared the different choices of semantic feature $F_i^s$ in $i$-th SDM. For the clarity, we list the main results in the table below, where $F_{i}$ means the output feature of stage $i$ in backbone, which is also the input feature of $i$-th SDM.
> It can be seen that:
> + The segmentation performance will be seriously degraded when the semantic feature of SDC input is implemented by the low-level feature $F_{1}$.
> + When using $F_{i}$ (i.e., the input feature) as semantic feature, the semantic feature has the same level semantics as the input feature, which may not exert the true power of SDC.
> + With the higher-level feature $F_{i+1}$ at the next scale $i+1$ with richer semantics as the semantic feature, the performance achieves the best.
>
> |   Semantic feature $F_i^s$   |     mIoU     |
> | :----: | :----: |
> |  $F_1$       | 79.3          |
> |   $F_i$       | 82.2          |
> |   $F_{i+1}$       | 82.9          |

---

> ### Author Response · Authors · 2022-08-07
> **Thanks for the comments from Reviewer XkWA !**
>
> Dear reviewer XkWA:
>
> Hello!
>
> We are really honored by your appreciation! Please allow us to express our sincere thanks!
>
> In our response comments, we have carried out targeted explanations and experiments for your suggestions and concerns, including the further explanation of table 5(a), the comparison of SDC and CDC, and the further analysis of the semantic difference term.
>
> We would like to hear from you further. If you have any questions, please feel free to let us know. We will try our best to solve your concerns.
>
> Best regards.
> The authors of “Semantic Difference Convolution for Semantic Segmentation”.

---

### Official Review · Reviewer_GLui · 2022-07-11

**Rating:** 4
**Confidence:** 4
**Soundness:** 3 good
**Presentation:** 3 good
**Contribution:** 2 fair

**Summary:**

This paper proposes a novel convolution-based operation/operator that could utilize the semantic features in an efficient and effective way for the segmentation task. The main contributions of this paper are (1) a new operation (semantic difference convolution, SDC) that is effective in the segmentation of target objects with precise boundaries, (2) the related module (SDM) to enhance the boundary information at the feature level, and (3) the proposed network exhibit the state-of-the-art semantic segmentation performance with precise boundaries of the target objects.

**Questions:**

* Questions & Discussions
1. The semantic distance (SD in equation 3) should be discussed. The SD is calculated using the Euclidean distance of two semantic features (f) with different positions. Intuitively, SD value by similar feature exhibits small value, whereas SD value exhibits magnified values when calculating boundaries. At this point, the reviewer is curious about the followings:

- 1-1. The authors’ address is that f exhibits semantic feature, and the Euclidian distance between two f exhibits semantic distance. Here, the definition and the mathematical illustration for the semantic feature and its meaning should be discussed, or a reference is required.

- 1-2. The normalization method (even without) can extremely suppress the values of the inner parts of the objects. In general, the color and semantic features significantly exhibit similar values in the same objects. Therefore, the SD value inside the object can be almost 0. Whereas the SD values nearby boundaries could be extremely magnified. Therefore, the reviewers worried that extremely small inner values degrade feature extraction for semantic features related to the target objects and only focus on the boundary-oriented features, and thus the predicted segmentation mask includes big halls inside.

2. Since the convolution operation can be utilized in many applications, the reviewer is curious that the motivation of the proposed operation in the semantic segmentation task. The proposed operation can be used in general applications such as classification and image generation tasks.


**Ethics Review Area:**

["I don’t know"]

**Limitations:**

1. Major issues
- The limited novelty should be discussed. As illustrated above (Question), the proposed operation exhibits limited novelty. Despite the simple yet effective mathematical illustration of the proposed convolutional operation, the strict mathematical modeling for the operation and mathematical analysis of why the proposed operation exhibits outstanding performance in segmenting objects’ boundaries should be discussed.

- The experiments should be improved. As the reviewer already understands, the evaluation metric of “mean Intersection over Union (mIoU)” can qualitatively measure the object details. However, recent studies [1-3] proposed new evaluation metrics to measure the object detail (especially boundaries of the target objects) quantitatively. Otherwise, the visualization of the feature map could illustrate the novel feature extraction when processing object details. Please refer to the activation maps [4]. To clear the authors’ addresses, in terms of the improved boundary-oriented segmentation, more experimental or mathematical evidence should be justified.

- Since the authors proposed the convolution-based operation, the mathematical modeling of the operation should be discussed with the parameters used for the previous convolution operation. The mathematical modeling can include the rank, dimension, size, and inputs (parameters). For instance, the size of the output feature map can be determined as --- when the padding parameter is “valid” or “same”, or when using dilation parameters. The author should illustrate the mathematical form to calculate the output shape using kernel size, padding, and strides (additionally dilation). Please refer to this page [5] for the mathematical form.


[1] Fernandez-Moral, Eduardo, et al. "A new metric for evaluating semantic segmentation: leveraging global and contour accuracy." 2018 IEEE intelligent vehicles symposium (iv). IEEE, 2018.

[2] Lee, Kyungsu, et al. "Boundary-oriented binary building segmentation model with two scheme learning for aerial images." IEEE Transactions on Geoscience and Remote Sensing 60 (2021): 1-17.

[3] Cheng, Bowen, et al. "Boundary IoU: Improving object-centric image segmentation evaluation." Proceedings of the IEEE/CVF Conference on Computer Vision and Pattern Recognition. 2021.

[4] Selvaraju, Ramprasaath R., et al. "Grad-cam: Visual explanations from deep networks via gradient-based localization." Proceedings of the IEEE international conference on computer vision. 2017.

[5] https://www.tensorflow.org/api_docs/python/tf/nn/conv2d

2. Minor issues
- The reviewer recommends reviewing the grammar and typo errors to improve the quality of the manuscript.


**Strengths And Weaknesses:**


1. Strengths
- The reviewer significantly understands the significance and importance of the task proposed in this paper. Segmenting various objects in real-world images is significantly important for many applications. Especially, clear and precise boundaries are required for the precise semantic segmentation task.

- Reproducibility of the manuscript. The manuscript is well organized in terms of exhibiting the hyper-parameters and model architecture.

- The manuscript is well organized and well written.

2. Weakness
- More detailed descriptions are illustrated in the “Question” and “Limitation” sections. Please see below.

---

> ### Author Response · Authors · 2022-08-02
> **Response to ''Questions'' comments**
>
> Thank you very much for your patient review. We will try our best to address all your concerns, and we sincerely hope to get your appreciation of this work. If you feel that your concerns have been better addressed, we sincerely hope the score could be improved.
>
> ---
>
>  **Q1-1. About the mathematical illustration, reference, and further discussion about the semantic feature $f$ in SDC.**
>
> Thank you for your suggestion, and we will add this to the revised version.
>
> Considering a parameterized non-linear deep learning model $F_{\theta}(\cdot)$ mapping an image $I$ to high-order pixel-wise feature maps $X$. The network contains $L$ stacked differentiable neural layers, we denote the output features from the $\ell$-th layer as $X^\ell$. Consider the index of two different layers $\ell_1$ and $\ell_2$  , where $\ell_2 > \ell_1$. An academic consensus [r1_1,r1_2, r1_3] is that with the deepening of the number of layers and the expansion of the cumulative receptive field, features will become more abstract and high-level, and more attention will be paid to category-related (object-level) semantic information, while the lower-level (high resolution) features contain more texture and detail [r1_1]. \
> SDC takes the higher-level features $X^{\ell_2}$ as the semantic feature as the guidance to enhance the boundary of low-level feature $X^{\ell_1}$ from the $\ell_1$-th layer, which blurs textures (false boundary) of the inner-Object while enhancing the real boundary between different categories.
>
> + [r1_1] Gedas Bertasius, Jianbo Shi, Lorenzo Torresani. High-for-Low and Low-for-High: Efficient Boundary Detection from Deep Object Features and its Applications to High-Level Vision. ICCV 2015.
> + [r1_2] Young-Yi Lin, Piotr Dollar, Ross Girshick, Kaiming He, Bharath Hariharan, Serge Belongie. Feature Pyramid Networks for Object Detection. CVPR 2017.
> + [r1_3] Jie Xu, Huayi Tang, Yazhou Red, Liang Peng, Xiaofeng Zhu, Lifang He. Multi-level Feature Learning for Contrastive Multi-view Clustering. CVPR 2022.
>
> ---
>
> **Q1-2. Concerns about the impact when the SD value is minimal.**
>
> This is a valuable question!
>
> As we expect, the SD value of SDC will indeed reach a very small value inside the object, which can effectively filter out the false boundaries caused by the inner-object texture. Because the role of SDC is to highlight category boundaries, the features of SDC cannot be directly used to produce segmentation results. Therefore, we have further designed a lightweight and efficient SDM module that flexibly combines the SDC features with the features from the backbone, avoiding the problem that the internal features of the object are completely suppressed. We illustrate this with the following experiments on Cityscapes-Val.
>
> | Feature fed into the decoder  |  $~~~$  mIoU  |
> | :---- | :----: |
> | Backbone-feature | 81.1 |
>  |    SDC-feature  |  61.4  |
>  | SDM-feature | 82.9 (+1.8)   |
>
>
> When using OCRNet (HRNet48) as the baseline, the final performance is significantly degraded if only SDC features are used as input to the decoder. After SDM fuses SDC features with Backbone's features, the model's performance significantly exceeds baseline.
>
> This could be also demonstrated in Figure 3 of the supplementary material. The SDM module fuses SDC features with the backbone features flexibly, which not only preserves clear boundaries but also avoids excessive suppression of internal features of objects.
>
> ---
>
> **Q2. The motivation of SDC and further applications.**
>
> + **Motivation:** The motivation is very intuitive, that is, we find that the vanilla convolution operator will cause local features to be blurred and smooth,  which makes it difficult for CNN to model fine boundary information. This weakness is evident in the semantic segmentation task. In this paper, we propose a new operator that can filter out the pseudo-inner-object boundary and enhance the real category boundary, significantly alleviating the problem of blurred boundaries caused by Vanilla-Conv.
>
> + **Further applications:** In the future, we plan to extend SDC/SDM to other visual topics, such as instance segmentation, panoptic segmentation, image super-resolution, and image denoising.

---

> > ### Comment · Reviewer_GLui · 2022-08-09
> > **Response to Rebuttal**
> >
> > * I would like to express my appreciation to the authors to take the time to my questions. Most of my concerns were resolved in the rebuttal phase; mainly related to the limitations and novelty I have addressed. The author's rebuttal to my questions and other reviewers' questions have resolved my misunderstanding and I could understand this paper in detail. In addition, the newly exhibited experimental result could express the empirical analysis of the proposed operator. Therefore, I will increase my rating through the discussion phase.
> >
> > * Furthermore, for reproducibility and the improvement of the deep learning society, I would strongly address that the code would be published in public. In hopeful expectation, I will adjust my rating in good faith.

---

> > > ### Author Response · Authors · 2022-08-09
> > > **Thanks for Reviewer GLui !**
> > >
> > > Dear Reviewer GLui:
> > >
> > > Sincerely thanks for your final appreciation of our work!
> > > It's really good news that your concerns have been addressed during this discussion phase.
> > > We will improve the final version of this paper according to the valuable suggestions from reviewers.
> > > Actually, our method is easy to follow. To ensure the reproducibility, the complete code, detailed config files and trained models will be public.
> > > Sincerely thanks and expect your higher rating!
> > >
> > > Best regards!  \
> > > The authors of “Semantic Difference Convolution for Semantic Segmentation”.

---

> > > ### Author Response · Authors · 2022-08-10
> > > **Response to Reviewer's comments.**
> > >
> > > Dear reviewer:
> > >
> > > We feel very honored and glad to hear from you! Your comments have helped our paper become stronger. I would like to extend my sincere thanks to you!
> > >
> > > **About the code:** The codebase involved in this paper, training config, and model checkpoint will be public!
> > >
> > > **About rating score:** We notice that the rating score is still 4 (Borderline reject). Since the rebuttal deadline is approaching, we are wondering that is there any other concerns. If so, please feel free to let us know and we will try our best to address your concerns! Honestly, we really wish to get a higher rating score from you.
> > >
> > > Best regards
> > >
> > > Authors

---

> ### Author Response · Authors · 2022-08-02
> **Response to ''Limitations'' comments (part-1)**
>
> Thank you for your comments! We will add the relevant contents you mentioned to the appendix according to your suggestions, including theoretical analysis of performance, visualization based on grad cam, and performance under new evaluation metrics.
> If you are satisfied with our reply, we sincerely hope you can improve your score.
>
> **1: Further mathematical analysis of the superior performance of the proposed method.**
>
> Here, we will combine the physical model of the classical anisotropic diffusion to illustrate our method theoretically. (**Please check the revised supplementary material for details**)
>
> Considering a parameterized deep model $F_{\theta}(\cdot)$ mapping an image $I$ to pixel-wise feature maps $X$. The network contains $L$ stacked neural layers, we denote the features from the $\ell$-th layer as $X^\ell$. Consider the index of two layers $\ell_2 > \ell_1$. An academic consensus [r1_1,r1_2, r1_3] is that with the deepening of the number of layers and the expansion of the cumulative receptive field, features will become more abstract and high-level, and more attention will be paid to category-related (object-level) semantic information.
> While the traditional convolution operator fuses local features, it also indiscriminately blurs the boundary information of local details, which is the inherent reason for the fuzzy prediction of the deep model at the boundary. Therefore, an intuitive idea is to improve the Conv operator to make it have local perceptual anisotropy.
>
> To this end, we turn to the following partial differential equation of the anisotropic diffusion process:
>
> $
> \partial u_t / \partial t = \text{div}(g(\nabla f) \nabla u_t),
> $
> $
> u_0 = x^{\ell_1},
> $
>
> where $f = x^{\ell_2}$ refers to the higher-level semantic feature of the input, which is fixed to the current diffusion process. $u_0=x^{\ell_1}$ represents the lower-level feature that we want to refine, and $f$ and $u$ maintain the same width and height by interpolation.
> $t$ is the time step, $\text{div}$ is the divergence operator and $\nabla$ is the gradient operator.
> $g(\cdot)$ is the so-called diffusion coefficient function, which is sensitive to changes (gradient) in local semantic features.
> The above equation is based on the theory that mass is conserved during diffusion, rather than being created and destroyed. Similar diffusion models are widely used in traditional image filtering theory [r1_4, r1_5].
>
> The physical meaning described by this partial differential equation is that the diffusion in places where the local high-level information changes violently (more likely at the category boundary) should be treated with caution (to avoid the boundary being blurred). For places where the high-level information changes gently, the diffusion process should be smooth (the false boundary caused by the texture inside the object should be blurred).
>
> This partial differential equation can be approximately solved by the finite difference method, for example:
>
> $u_{t+1}[h, w] = u_{t}[h, w] + \delta,
> $
> $$
> \text{where } \delta = g(f[h-1, w] - f[h, w]) * (u_{t}[h-1, w] - u_{t}[h, w]) \\ + g(f[h+1, w] - f[h, w]) * (u_{t}[h+1, w] - u_{t}[h, w]) +g(f[h, w-1] - f[h, w]) * (u_{t}[h, w] - u_{t}[h, w])  +g(f[h, w+1] - f[h, w]) * (u_{t}[h, w+1] - u_{t}[h, w])
> $$
>
> However, the stability and convergence of the finite difference method require the fine setting of boundary conditions and related parameters, and its iterative operation process will bring significant time and memory consumption, especially in the training phase, which is easy to cause numerical instability.
>
> Our SDC/SDM uses a learnable parameterized mapping to model the PDE solving process (**Please check the revised supplementary material for details**), and a similar method of learning PDE solution through the neural network can be found in [r1_6].
>
> We also compared our SDC/SDM with the finite difference method (iterative to convergence, 20 runs to take the average performance) to process the backbone's features in the semantic segmentation model. Our learnable procedure is much better and significantly more stable than the finite difference method. This further explains the superiority of our method.
>
>
> | Solver    | $~~~~$mIoU |
> | :---- | :----: |
> | Finite difference method | 80.2 $\pm$ 1.4 |
> | Our  SDC/SDM |  82.9 $\pm$ 0.2|
>
>
>
>
>
>
>
> [r1_4] Pietro Perona and Jitendra Malik.  Scale-space and edge detection using anisotropic diffusion. T-PAMI 1990.
> [r1_5] Guillermo Sapiro. Geometric partial differential equations and image analysis. Cambridge University Press.
> [r1_6] Johannes Brandstetter, Daniel Worrall, Max Welling. Message Passing Neural PDE Solvers. ICLR 2022.
> [r1_7] Yuehaw Khoo, Jianfeng Lu, Lexing Ying. Solving parametric PDE problems with artificial neural networks. EJAM.

---

> ### Author Response · Authors · 2022-08-02
> **Response to ''Limitations'' comments (part-2)**
>
> **2: About the novelty.**
>
> Thank you for your recognition of the simplicity and effectiveness of our work. \
> Here, we discuss in detail the contributions  and novelty in this paper.
> + This paper analyzes for the first time the reason why it is difficult to model fine boundaries in deep networks, that is, the vanilla-convolution operator tends to blur the local details, and the stacked multiple convolutional layers aggravate the ambiguity problem, which is the intrinsic cause of the difficulty in modeling the boundary details in deep models.
> + For the intrinsic reasons of the blurred boundary information in the deep model, we design a very simple and efficient boundary enhancement operator, termed SDC, which can strengthen the real object boundary.
> + In order to make our method easily fit with various deep segmentation models, we designed a flexible and lightweight module, SDM, based on SDC. This module can be inserted into most existing deep segmentation models as a neck part, without modifying the original network structure.
> + Our approach significantly improves the segmentation performance (mIoU, and Boundary F-Score) of the baseline models on multiple current public datasets.
>
>
> **3: Further evaluation of new metrics.**
>
> Thank you for your comments. In our paper, we have used mIoU and Boundary F-score to measure the performance of the model at the object level and Boundary detail level respectively, see Table 1 and Table 2 for mIoU, Table 3 and Table 5 for the Boundary F-score.
>
>
> Fernandez-Moral. [1] proposed a new metric that accounts for both global and contour accuracy in a simple formulation.
> Kyungsu Lee [2] proposed a new metric, which is the boundary-oriented intersection over union B-IoU for quantitative evaluation of the shapes and boundaries of the model.
> Bowen Chen [3] proposed the Boundary IoU (Intersection-over-Union), a new segmentation evaluation measure focused on boundary quality. We choose [3], the latest of the three, as the evaluation metric to compare the baseline (HRNet48-OCRNet) and ours (baseline + SDM). The performance is as follows. Clearly, our SDM significantly improves the performance of the Baseline model on the boundary metric [3].
>   Method     |  Boundary IoU
> | ---- | :----: |
> | Baseline      | 62.4    |
> | Baseline  + SDM    | **66.1 (+3.7)**      |
>
> [1] Fernandez-Moral, Eduardo, et al. "A new metric for evaluating semantic segmentation: leveraging global and contour accuracy." 2018 IEEE intelligent vehicles symposium (iv). IEEE, 2018.
> [2] Lee, Kyungsu, et al. "Boundary-oriented binary building segmentation model with two scheme learning for aerial images." IEEE Transactions on Geoscience and Remote Sensing 60 (2021): 1-17.
> [3] Cheng, Bowen, et al. "Boundary IoU: Improving object-centric image segmentation evaluation." Proceedings of the IEEE/CVF Conference on Computer Vision and Pattern Recognition. 2021.
>
>
>
> **4: Further visualization (Grad-CAM).**
>
> Thank you for your comments. **Please check Section E.3 of the revised supplementary material for the Grad-CAM visualization comparison between the baseline model (HRNet48-OCRNet) and ours (baseline + SDM).**
> When performing the Grad-CAM visualization, we set the feature map fed into the decoder as the target layer's feature map.
> It is clear that SDM makes the activation region of the model more concentrated inside the object, and the boundary of the activation region is more consistent with the real semantic boundary of the object.

---

> ### Author Response · Authors · 2022-08-02
> **Response to ''Limitations'' comments (part-3)**
>
> -------------
>
>
> **5: Further discussion with the parameters in the SDC operator (input and output size, padding, dilation, kernel-size, strides…).**
>
> Thank you for your advice. Here, we refer to Page [5] to discuss the parameters involved in the use of SDC, which we will add to the supplementary materials.
>
> SDC takes two 4-D tensor as input, namely the feature map $X \in \mathcal{R}^{B \times C_i \times H \times W}$ and the semantic feature map $F \in \mathcal{R}^{B \times C_f \times H \times W}$. Parameters such as kernel-size in SDC have the same meaning as those in vanilla convolution. Given a kernel tensor of shape $w \in \mathcal{R}^{C_o \times C_i \times h \times w}$, where $C_o, C_i, h, w$ is the out_channels, in_channels, filter_height, filter_width, this op performs the following:
> 1. Extract the feature patch $x \in \mathcal{R}^{B \times C_i \times h \times w}$ from $X$ and the semantic feature patch $f \in \mathcal{R}^{B \times C_f \times h \times w}$ from $F$, according the parameter settings (padding, strides, dilation, kernel-size).
> 2. Calculate the central difference map $d \in \mathcal{R}^{B \times C_i \times h \times w}$ via $d[b, c, h, w] = x[b, c, h, w]  - x[b, c, h_c, w_c] $. Then, flatten d to shape $B \times C_i h w$
> 3. Calculate the semantic difference map $s \in \mathcal{R}^{B \times 1 \times h \times w}$ via $d[b, c, h, w] = x[b, c, h, w]  - x[b, c, h_c, w_c] $. Then, repeat $C_i$ times for $s$ at the dimension-1 and flatten it by following 1, thus, we get the $s$ matrix of shape $B \times C_i h w$.
> 4. Flatten the kernel tensor $w$ to a matrix of shape $ C_i  h w \times C_o$.
> 5. Performing the the element-wise multiplication and matrix multiplication to get the output matrix $o \in \mathcal{R}^{B \times C_o}$ via $o = (d \cdot s) \otimes  w$.
> 6. Reshape $o$ from $B \times C_o$ to $B \times C_o \times 1 \times 1$ via as the output tensor of the current location.
>
> Thus, given the input feature maps of hight/width of $H/W$, the dilation-rate of $d$, the kernel-width/height of $k$, the stride of $s$, the padding size of $p$, the height/width ($H_o/W_o$) of the output feature map should be  $H_o= [H + 2 p - d(k-1) - 1]/s+ 1$, and $W_o = [(W + 2p - d(k-1) - 1)/s] + 1$.

---

> ### Author Response · Authors · 2022-08-07
> **Looking forward to further comments from Reviewer GLui.**
>
> Dear reviewer Glui:
>
> Hello.
>
> Thank you for your review of this article and your contribution to the academic community! Thanks for your constructive suggestions! In our response comments, we made a detailed experimental and theoretical explanation for your questions and concerns, including:
>
> A. In [Response to ''Questions'' comments](https://openreview.net/forum?id=mmzkqUKNVm&noteId=PbmvXh5ylsT2), we made the following responses: (1). further mathematical illustration, reference, and further discussion about the semantic feature in SDC; (2). Further analysis and experiments of the impact when the SD value is minimal; (3). The motivation of SDC and further applications.
>
> B. In [Response to ''Limitations'' comments (part-1)](https://openreview.net/forum?id=mmzkqUKNVm&noteId=Y75Q0hcLXEG), we provided a further mathematical (theoretical) analysis of the performance of the proposed method.
>
> C. In [Response to ''Limitations'' comments (part-2)](https://openreview.net/forum?id=mmzkqUKNVm&noteId=aup8-JJn_Z0), we further clarified the novelty, provided the performance under the new metric, and further Grad-CAM based visualization.
>
> D. In  [Response to ''Limitations'' comments (part-3)](https://openreview.net/forum?id=mmzkqUKNVm&noteId=stKc-4GCeQWN), we provided further discussion with the parameters in the SDC operator (input and output size, padding, dilation, kernel-size, strides…)
>
> &nbsp;
>
> We would like to know whether our reply has addressed your concerns. If you have any questions, please feel free to let us know, and we will try our best to address your concerns. We sincerely hope to get recognition from you and even the academic community. If our reply makes you feel satisfied, we also sincerely hope to get a higher rating score from you.
>
> &nbsp;
>
> Best regards! \
> The authors of “Semantic Difference Convolution for Semantic Segmentation”

---

> ### Author Response · Authors · 2022-08-08
> **Looking forward to the reviewer's reply !**
>
> Dear reviewers,
>
> Thanks a lot for your time and efforts in reviewing our paper. We have tried our best to address all mentioned concerns. We would appreciate it if you could take a look at our response. As the discussion deadline is approaching, your feedback is very important to us, and if there are any new questions, we can therefore reply in time.
>
> sincerely yours
> Authors

---

### Meta-Review · Area_Chair_k5pM · 2022-08-24

**Recommendation:** Accept
**Confidence:** Certain

**Metareview:**

This submission got a mixed rating: 1 borderline reject, 2 week accept and 1 accept.

Most of the concerns lie in the explanations on the details and experimental comparison with certain baselines/variants. The authors addressed them well by providing additional experiment results in their response.

The remained concern from the reviewers giving borderline reject lies in theoretical justification of the proposed operation. The authors managed to provide a theoretical interpretation from the viewpoint of diffusion process, which partially addresses the reviewer's question.

Overall, all the reviewers agree that this submission introduces a simple and effective method for the segmentation field. The effectiveness of the proposed method has been validated via extensive experiments. The performance improvement is significant. The manuscript is written clearly. The contribution is sufficient.

Based on the above considerations, AC recommends accept for this submission.

**Award:**

No

---

### Decision · Program_Chairs · 2022-09-14

Accept